# Comparative Analysis of Circular and Square End Plates for a Highly Pressurized Proton Exchange Membrane Water Electrolysis Stack

**Minjeong Jo [1], Hyun-Seok Cho [2] and Youngseung Na [1],***

[1]  Department of Mechanical and Information Engineering, University of Seoul, Seoul 02504, Korea; jmj4521@uos.ac.kr

[2]  Hydrogen Research Department, Korea Institute of Energy Research, Daejeon 34101, Korea; hscho@kier.re.kr

*  Correspondence: ysna@uos.ac.kr; Tel.: +82-(0)2-6490-2395

**Abstract:** End plates are located at both ends of a proton exchange membrane water electrolysis (PEMWE) stack. If the end plates are thin, clamping pressure is not uniform and the performance of PEMWE can deteriorate from leakage and high electrical contact resistance caused by the deformation of the thin end plates. In this study, end plates were designed to reduce the weight while clamping the stack uniformly by finite element analysis (FEA). The weights of the circular and square end plates were reduced compared to conventional end plates by 22.9% and 23.3%, respectively. The stress and strain distribution of square and circular end plates are analyzed using topology optimization. This analysis can improve the performance of the PEMWE by using new end plate designs verified by dummy cell stack simulation to maintain uniform pressure.

**Keywords:** PEMWE; end plate; finite element analysis; weight reduction; pressure uniformity; topology optimization

## 1. Introduction

Air pollution, carbon dioxide emissions, and fine dust problems have attracted attention due to the use of fossil fuels. Proton exchange membrane water electrolysis (PEMWE) can produce hydrogen without emitting pollutants such as carbon dioxide, in contrast to fossil fuels [1]. If electricity is produced by renewable energy technology, hydrogen from water electrolysis can be stored in an environmentally friendly manner [2]. PEMWE consists of an end plate, current collector, bipolar plate (BP), porous transport layer (PTL), and membrane electrode assembly (MEA) [3]. The end plates—ones of the components of the PEMWE—are located at both ends of the stack to maintain a constant pressure for lower electrical resistance and tight gas sealing [4]. Thick end plates are produced to withstand the high pressure, resulting in heavy weight and unnecessary material consumption. In contrast, using thin plates can produce non-uniform clamping pressure, thus deteriorating the performance of PEMWE due to leakage and large electrical contact resistance [5].

Chang et al. [6] examined the effect of clamping pressure on the performance of a proton exchange membrane fuel cell (PEMFC). Clamping pressure can reduce the interfacial resistance between the BP and the gas diffusion layer (GDL). This study focuses on fuel cell performance based on the diffusion path for mass transfer in the GDL, not in the end plates. Furthermore, Wen et al. [7] conducted an experimental study of the clamping effects on the performance of a single PEMFC, and a 10-cell stack was studied with a pressure-sensitive film. The linear relationship of combinations of bolts and clamping torque was investigated experimentally. The pressure distribution in a 10-cell stack was not simulated in detail.

Wang et al. [5] conducted an experimental study on clamping pressure distribution with performance tests and pressure-sensitive films. Newly designed end plates provide uniform stress to MEA. The new design was convenient to use in the laboratory for the uniform clamping pressure, but not in practice because of its bulky system with the built-in hydraulically pressurizing devices. Bates et al. [8] applied an endplate with center holes and hex-head screws in single-cell and 16-cell stacks. Simulation and experimental analyses of single-cell and 16-cell stacks and components of the stack were performed by finite element analysis (FEA) and pressure-sensitive film. The stress on the GDL is simulated at 1.6 MPa for the 16-cell stack and 4 MPa in single-cell. Experimental results demonstrated nearly zero pressure at the center of the stack, regardless of the clamping pressure. Pushing pins through center holes compensate for the deficient clamping pressure of end plates, which needs additional devices for the stacks.

Lin et al. [9] investigated the multi-objective topology optimization of end plates with nonlinear contact boundary conditions. The 5-cell and 10-cell stacks have different response forces to end plates, which optimize the different shapes of end plates. The stiff end plates press the cells uniformly, but the parametric study of various end plates was not investigated further. Lee et al. [10] used FEA and a pressure film test to study end plate assembly pressure distribution. The quantitative comparison of simulation and experimental results are essential for obtaining consistent fuel cell performance with stacking parameters such as stacking design, BP thickness, sealing size, and assembly pressure. Asghar et al. [11] designed and manufactured end plates for a 5 kW PEMFC that provided uniform pressure distribution between the fuel cell components (e.g., BPs, GDLs, and current collectors) and, consequently, reduced the contact resistance, which was measured by electrochemical impedance spectroscopy. The thickness of the end plates was optimized, minimizing deflections. The clamping torque, end plate thickness, and number of bolts were optimally combined for the appropriate assembly pressure distribution. Montanini et al. stated that higher torque does not ensure the uniform pressure distribution because of the bending of end plates [12].

For PEMWE, Al Shakhshir et al. [3] studied the in situ experimental characterizations of clamping pressure effects. A graphite block is stiffer than a titanium block for loading uniform pressure to the MEAs, which results in homogeneous local current density [13]. Similar to the additional hydraulic compression in PEMFC, the pneumatical compression of the end plates in PEMWE stacks enhances the uniformity of local current density [14]. Non-uniform distribution of current density can result in a temperature difference and significant degradation of MEAs [15]. Wilson et al. [16] have an end plate design patent for a high-pressure electrolytic module that maintains a uniform pressure with the backing plate. Verdin et al. investigated the relationship between pressure distribution and local current density with the operando method [17]. The highly compressed region exhibits a high current density with low ohmic resistance. Selamet et al. compared circular PEMWE stacks with 5 and 10 cells [18].

The further analysis of clamping pressure distribution in the stacks with different cell numbers is necessary to investigate the performance difference between the stacks. Although the previous studies conducted PEMFC and PEMWE end plate research, few papers have studied high pressure sealing end plates with weight reduction. Comparing the square and the circular end plates is necessary to ensure the uniform pressure distribution at each cell in the stacks.

In this study, stress, deformation analysis, and topology optimization are performed to reduce the weight of circular and square end plates. After the topology optimization, design parameters, such as removing the regions of upper, lower, or side parts, were derived. Stress and deformation analyses were conducted with newly designed end plates. Strain simulation and validating experiments of dummy cells was performed to verify clamping pressure uniformity between cells by insertion of pressure sensing films into the middle and end cells of the stack. The square end plate experiences more stress than the circular end plate in topology optimization with weight reduction. In contrast, the clamping pressure between cells in the square end plate stacks is uniform similarly to cells of the circular end plate stacks based on FEA.

## 2. Modeling

### 2.1. Topology Optimization and Parameter Derivation

An analysis of the end plate pressure distribution and deformation point performed (using commercial simulation software ANSYS 18.1) to examine reducing the end plate weight. The clamping equipment consisted of bolts, end plates, and current collectors. The number of meshes converged to an average of 27,000 with less than 1% of the relative error between the element and deformation average curve. The boundary condition was applied to the head of all bolts combined at the top of the end plate, as depicted in Figure 1.

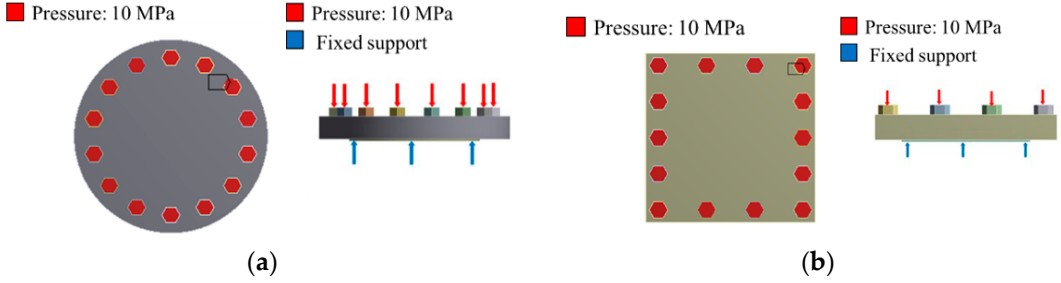

**Figure 1.** Pressure and fixed support boundary conditions for the (**a**) circular and (**b**) square end plates.

A force of 720,000 N, which is equivalent to the pressure of 10 MPa on the current collector area, was loaded on 14 bolts in the circular and square end plate simulations, respectively. A fixed support was applied to the base side of the current collector under the end plate to restrict the degrees of freedom of points, edges, and faces and for the transition. Table 1 lists the properties of the clamping equipment components.

**Table 1.** Material properties of simplified proton exchange membrane water electrolysis (PEMWE) components.

| Component | End Plate | Current Collector | Dummy Cell |
|:---:|:---:|:---:|:---:|
| Material | SUS 316L | Ni | Paper gasket |
| Density (kg/m$^3$) | 8000 | 8500 | 1200 |
| Poisson's ratio | 0.27 | 0.1 | 0.3 |
| Young's Modulus (GPa) | 193 | 210 | 2 |

In topology optimization, the region to be optimized is controlled by defining the design and exclusion regions. The design region can be optimized, whereas the exclusion region is fixed to avoid deformation by the solver. Therefore, the exclusion region was applied to the bolt and current collector, while the design region was applied to the end plate—which was similar to the boundary condition [19]. Of the topology optimization options, the maximum number of iterations was set up to the default value of 500, which is repeated by as many times as the convergence or setting value. The minimum normalized density was set to the default value of 0.001, and the percent retained at the response constraints was set to 50% or less to set the weight. Furthermore, the convergence accuracy was set to 0.1%, which is the default value.

### 2.2. Pressure Distribution Analysis

The simplified PEMWE stack consists of bolts, nuts, end plates, current collectors, and 21 dummy cells [20], which are simplified by using plain paper gasket plates as dummy cells without any patterns. The dimensions of main stack components are listed in Table 2. Dummy cell simulation is appropriate for depicting strain distribution.

**Table 2.** Dimensions of the simplified PEMWE components.

| Component | Conventional End Plate | | Current Collector | | Dummy Cell | |
|---|---|---|---|---|---|---|
| | Circular | Square | Circular | Square | Circular | Square |
| Diameter (circular) or side length (square) (cm) | 44.28 | 37.84 | 30.28 | 26.84 | 30.28 | 26.84 |
| Area (cm$^2$) | 1539.9 | 1431.9 | 720.1 | 720.4 | 720.1 | 720.4 |
| Thickness (mm) | 40 | 40 | 3 | 3 | 5 | 5 |

The total number of mesh elements in the stack was established at a similar level of approximately 50,000. The force was calculated (to represent the stack boundary condition similar to the actual phenomenon) by applying a value of 51,429 N for the circular end plate and the square end plate to each bolt, respectively, identical to the boundary conditions at topology optimization, which loads 10 MPa on to the BP area uniformly, as depicted in Figure 2.

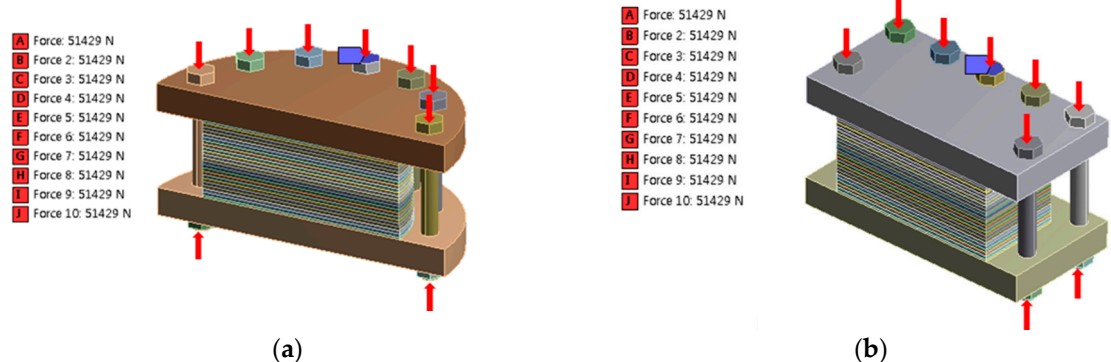

(**a**)                    (**b**)

**Figure 2.** Clamping pressure boundary conditions on (**a**) the circular stack and (**b**) the square stack.

## 3. Experimental Methods

Experiments were conducted at 1/4-length scale with conventional and newly designed end plates with paper gasket dummy cells to confirm the tendency of the dummy cell simulation. The geometry and materials of end plates are identical to the modeling parameters.

Torque is calculated by the equation from Al Shakshir et al. [14].

$$\tau = \frac{P_c \times A_c \times C \times D}{N},$$ (1)

where $P_c$ is the clamping pressure on the dummy cell area $A_c$. $C$ is the friction coefficient (0.2 for steel bolts). $D$ and $N$ are the nominal bolt diameter (inch) and the number of bolts, respectively. $\tau$ is the applied torque (inch-pounds). The calculated torque from Equation (1) was 34 lb·in for 14 bolts, respectively. The 21-cell stack was used for validation with the pressure sensing film (Fujifilm Prescale LW, Tokyo, Japan) at the end and the middle of the cells. The color scale of LW film ranged from 2.5 (white) to 10 MPa (red) according to the pressure.

## 4. Results and Discussion

### 4.1. Topology Optimization and Design Parameters

Figure 3 illustrates the results of the conventional end plate stress and deformation analysis. As depicted in Figure 3a,b, the maximum stress of the circular end plate was 62.248 MPa, the total deformation was 0.015193 mm, and the weight was 50.766 kg (end plate + current collector + bolt head). The maximum deformation occurred at the outer edge of the end plate due to the stress concentration in

the bolt. As depicted in Figure 3c,d, the maximum stress of the square end plate was 239.08 MPa, the total deformation was 0.029516 mm, and the weight was 47.308 kg (end plate + current collector + bolt heads). The maximum deformation occurred at each vertex of the end plate due to the stress concentration in the bolt.

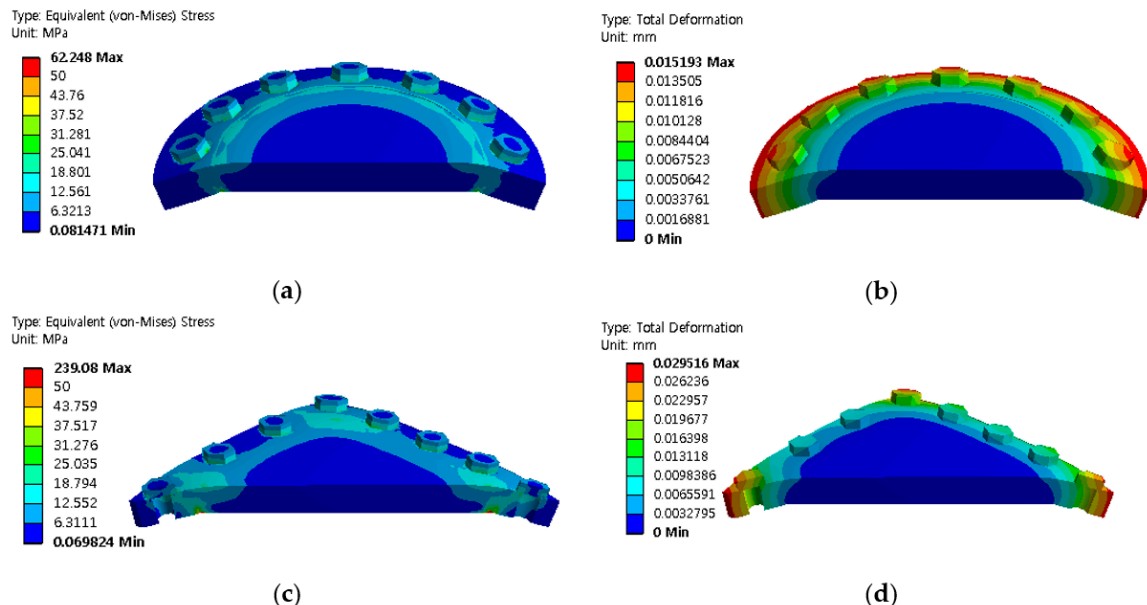

**Figure 3.** (**a**) Equivalent stress (MPa) and (**b**) total deformation (mm) of the conventional circular end plate; (**c**) equivalent stress (MPa) and (**d**) total deformation (mm) of the conventional square end plate.

From the topology optimization conducted using the Ansys 18.1 bundle, cut regions were suggested in each end plate: the side region between bolt locations (Figure 4a top), the groove of the bottom side (Figure 4a bottom), and the top center region (Figure 4c). For the topology region, whereas Figure 4a,c illustrates which exclusion region was applied only to bolts that were similar to the boundary condition, Figure 4b illustrates which exclusion region was applied to upper and lower sides of the end plate. Both sides remained after the topology optimization; therefore, the side-cut shape differs from that in Figure 4a.

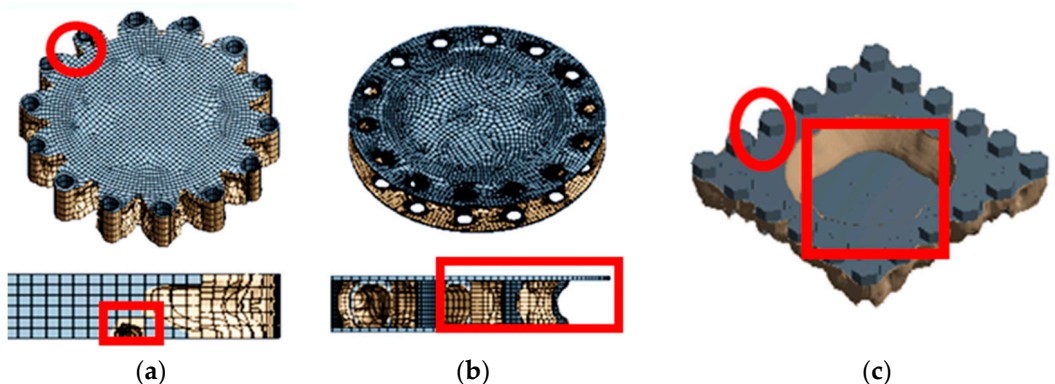

**Figure 4.** Topology optimization results of conventional end plate of (**a**,**b**) circular end plate and (**c**) square end plate.

Based on the topology optimization depicted in Figure 4, the four parameters were derived as follows: (1) volume removal between the bolts, (2) volume removal at the bottom, (3) round of sector form at the top, and (4) top center cut depth.

The designation of the end plates consists of six letters. The order of the removed parameters is as follows: the first letter from the left is the bolt, the second letter is the bottom side, the third and fourth digits are the round radius, and the fifth and sixth digits are the thickness. The first letter is the symbol "C" in Figure 4a with a red circle when completely removed, the symbol "I" with a red square at the bottom in Figure 4b when partially removed, and the symbol "N" when nothing has been removed. The second letter is the symbol "B" with a red square at the bottom left of Figure 4a, and the symbol "N" is the portion where nothing has been removed. Furthermore, the part where nothing has been removed for the third and fourth digits is indicated by the symbol "00", the part where a round R5 has been applied is indicated by the symbol "05", and the part where a round R15 has been applied is indicated by the symbol "15". The part where nothing has been removed for the fifth and sixth digits was designated "00", the portion to be removed to a depth of 10 mm was designated "10", and the portion to be removed to a depth of 20 mm was designated "20".

Based on the topology optimization of the square end plate depicted in Figure 4c, three parameters were derived: (1) volume removal between the bolts, (2) fillet formation, and (3) thickness reduction of the top region. Before proceeding with the parametric stress and strain analysis derived from the topology optimization of the square end plate, seven types were designed like the circular end plate, as illustrated in Figure 5b. The designed square end plate consists of four letters in total, in the order of removed parameters: the first letter from the left is the bolt, the second letter is the fillet, and the third and fourth digits are the thickness. The first letter is depicted in Figure 4c as a red circle, where the symbol "C" was completely removed, and the part where nothing was removed is indicated by the symbol "N". The second digit is indicated by the symbol "F" in the part where the symbol fillet is formed and the symbol "N" in the part where nothing is removed. Finally, the third and fourth digits are marked with the symbol "00" where nothing was removed, "10" where the part to be removed is 10 mm deep, and "20" where the part to be removed is 20 mm deep.

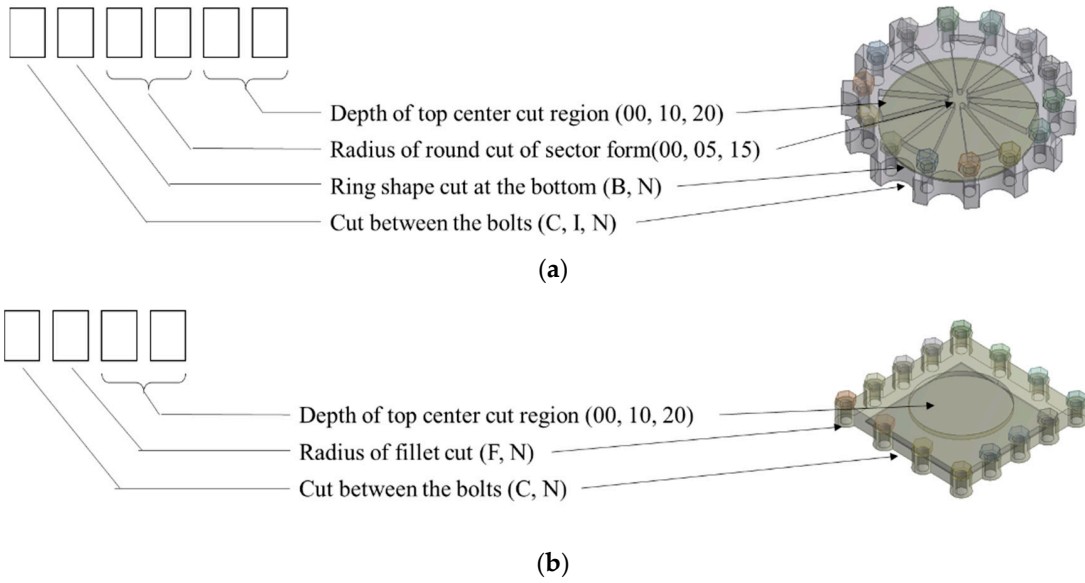

**Figure 5.** Code names of the parametric designs for the (**a**) circular and (**b**) square end plates.

## 4.2. Parametric Study of Various End Plates

As depicted in Table 3, NB0000 with the ring-type groove at the bottom of the end plate exhibits small deformation; it also exhibits small stress because of the removal of the concentrated stress region (i.e., the groove). For NN0510, the total deformation was increased by 23.5%, and the weight was decreased by 5.6%, which is a relatively greater weight reduction with a smaller total deformation. For CN0000, the total deformation increased by 14.7%, and the weight decreased by 17.4%, compared with the conventional circular end plate. The total deformation increased as the weight decreased;

CN0510 modeling was derived by adding CN0000, which had the lowest total deformation of the circular end plate parameters. For CN0510, the results imply a light circular end plate after confirming a 40.5% increase in total deformation and a 22.9% decrease in weight compared with the conventional circular end plate.

**Table 3.** Equivalent stress distribution and total deformation simulation results based on circular end plate geometry.

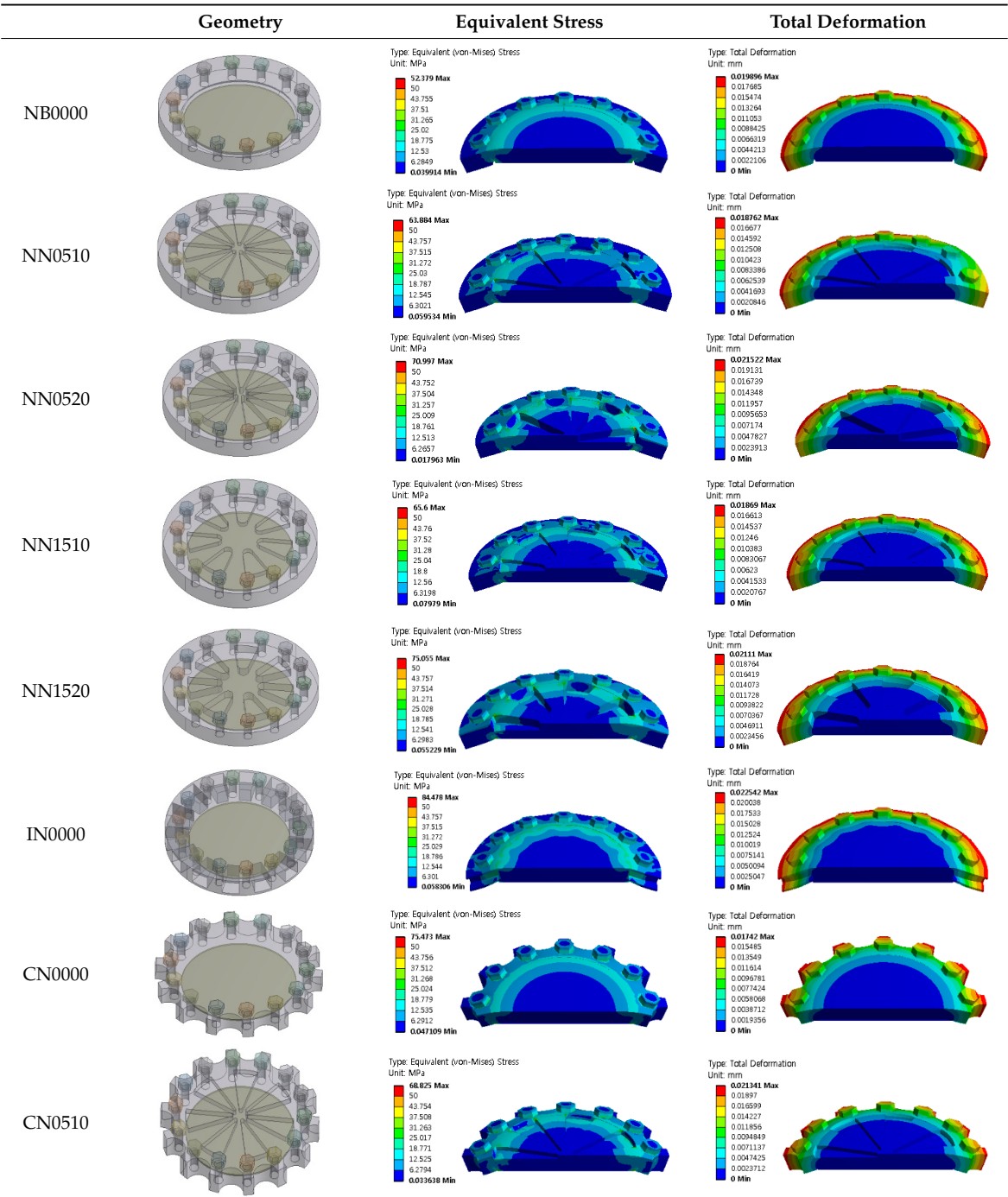

As depicted in Table 4, the stress and strain behaviors were confirmed by modeling the square end plate. The maximum deformation was observed at the outer edge of the end plate due to the stress concentration in the bolt, similar to the conventional square end plate. The stress of the square end

plates is much larger than that of the circular end plates by one order of the magnitude. The fillet of the end plates (NF00) does not affect the weight reduction significantly.

**Table 4.** Equivalent stress distribution and total deformation simulation results of the various geometrical type of square end plates.

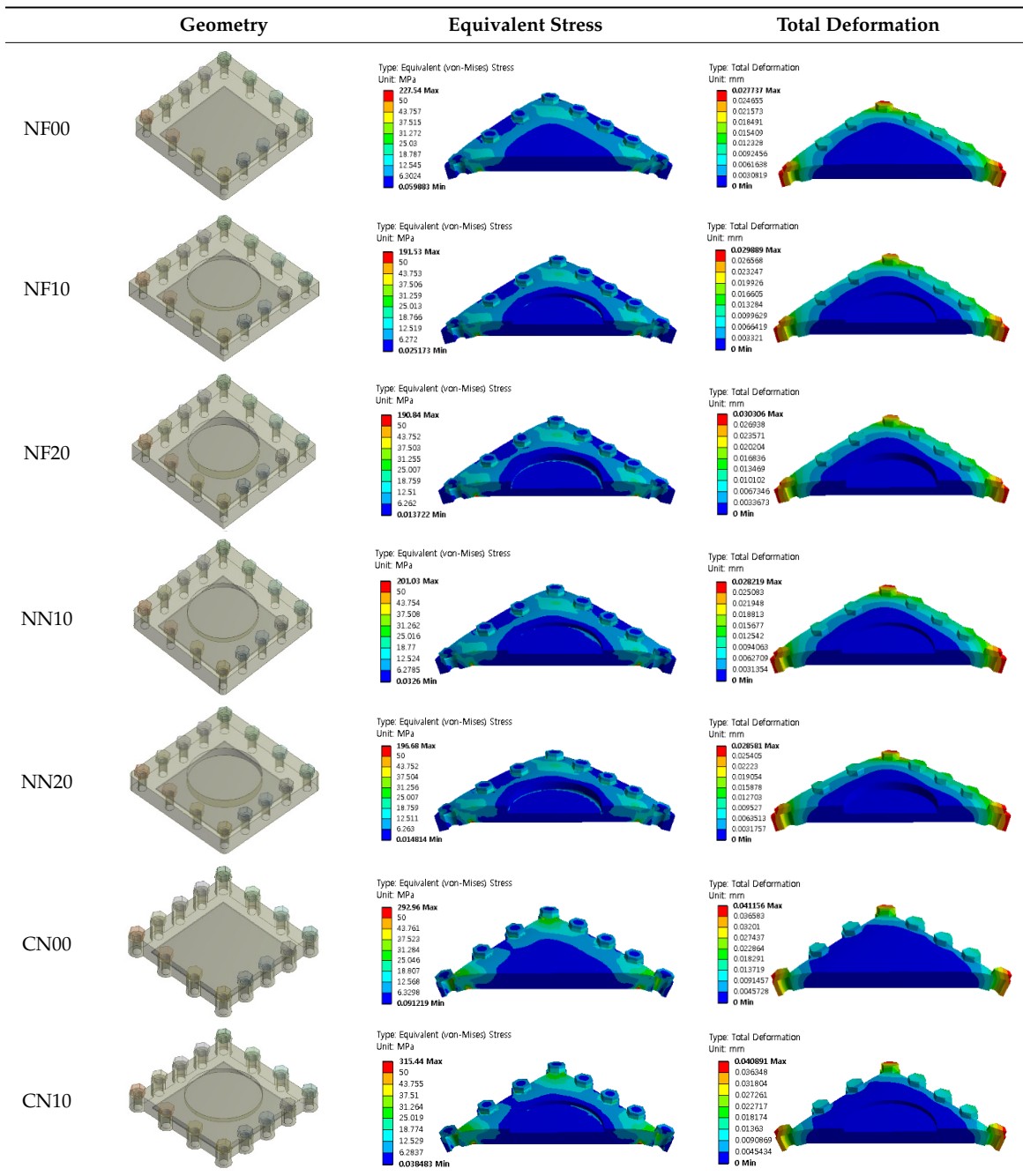

| | Geometry | Equivalent Stress | Total Deformation |
|---|---|---|---|
| NF00 | | | |
| NF10 | | | |
| NF20 | | | |
| NN10 | | | |
| NN20 | | | |
| CN00 | | | |
| CN10 | | | |

In contrast, NF10, NF20, NN10, and NN20 contribute to weight reduction from the center cut of the upper side of the end plates. For NN10, the total deformation decreased by 4.4%, and the weight decreased by 5.8%. For CN00, the total deformation increased by 39.4%, and the weight decreased by 17.5% compared with the conventional square end plate. The total deformation increased as the weight decreased; CN10 modeling was derived by adding CN00, which had the largest weight reduction among square end plate parameters. The results imply a light square end plate after confirming an increase of 38.5% and a decrease of 23.3% in weight, similar to the results in CN10.

For the circular end plate—as depicted in Figure 6a—when the weight of the circular end plate is reduced, the tendency of the removal portion can be confirmed. The effect of removing the outer edge of the end plate is the largest, and the effect of removing the bottom of the end plate is the smallest. Consequently, when the weight of the circular end plate is reduced, it becomes possible to derive a light end plate design with less deformation by removing it in the order of bottom, round, thickness, and outer edges. The weight reduction and strain data set have a linear relationship because the circular end plate, which is a symmetric rotating body, distributes the stress uniformly to the plate.

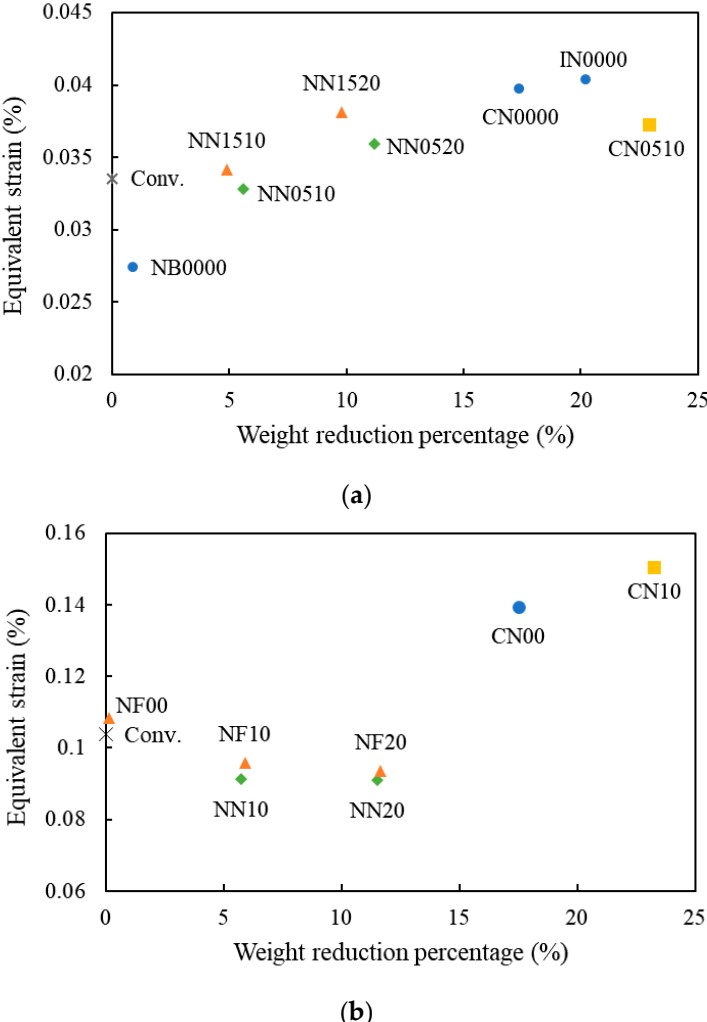

**Figure 6.** Weight and strain relationship of various end plate types for (**a**) circular and (**b**) square end plates.

In contrast to the circular end plate, in Figure 6b, the relationship between the strain and the weight reduction of the square end plates illustrates nonlinearity, which indicates the optimized combination of the weight reduction and the shape should be calculated on a case-by-case basis. The effect of removing the outer edge of the square end plate is the largest, and the effect of the removal parameter only applied to the end plate fillet is the smallest, as depicted in Figure 6b. Consequently, when the square end plate's weight is reduced, if the fillet, thickness, and outer edge are removed in order, the light end plate design can be derived with less deformation.

Furthermore, upon comparing the circular and square end plates as presented in Tables 3 and 4, the square end plates had greater stress concentration and deformation than the circular end plates. Moreover, as depicted in Figure 6b, the square end plates can deteriorate cells.

### 4.3. Dummy Cell Stack Simulation and Experimental Results

Table 5 illustrates the strains of the end and middle cells of the 21-cell stacks to which the conventional circular end plate and CN0510 end plate, which was optimized by the parametric study, were applied. The strain ranges are similar at the end and the middle cells of the stacks, respectively, between the conventional and CN0510 end plates. The wavy shape inside on the end cell occurred because the upper cut of the sector formed, which disappeared on the middle cell with CN0510 end plates. The outside of the cell was bent and appeared severely deformed with more than 0.001 of strain. Experimental results of pressure sensing films for circular end plate dummy cells have similar distribution of pressure. Both end plates pressed inside (pink) of the dummy cells lighter than outside (strong red).

**Table 5.** Comparison of the simulated equivalent-strain distribution with the experimental pressure distribution, using the circular end plates of the dummy-cell stack.

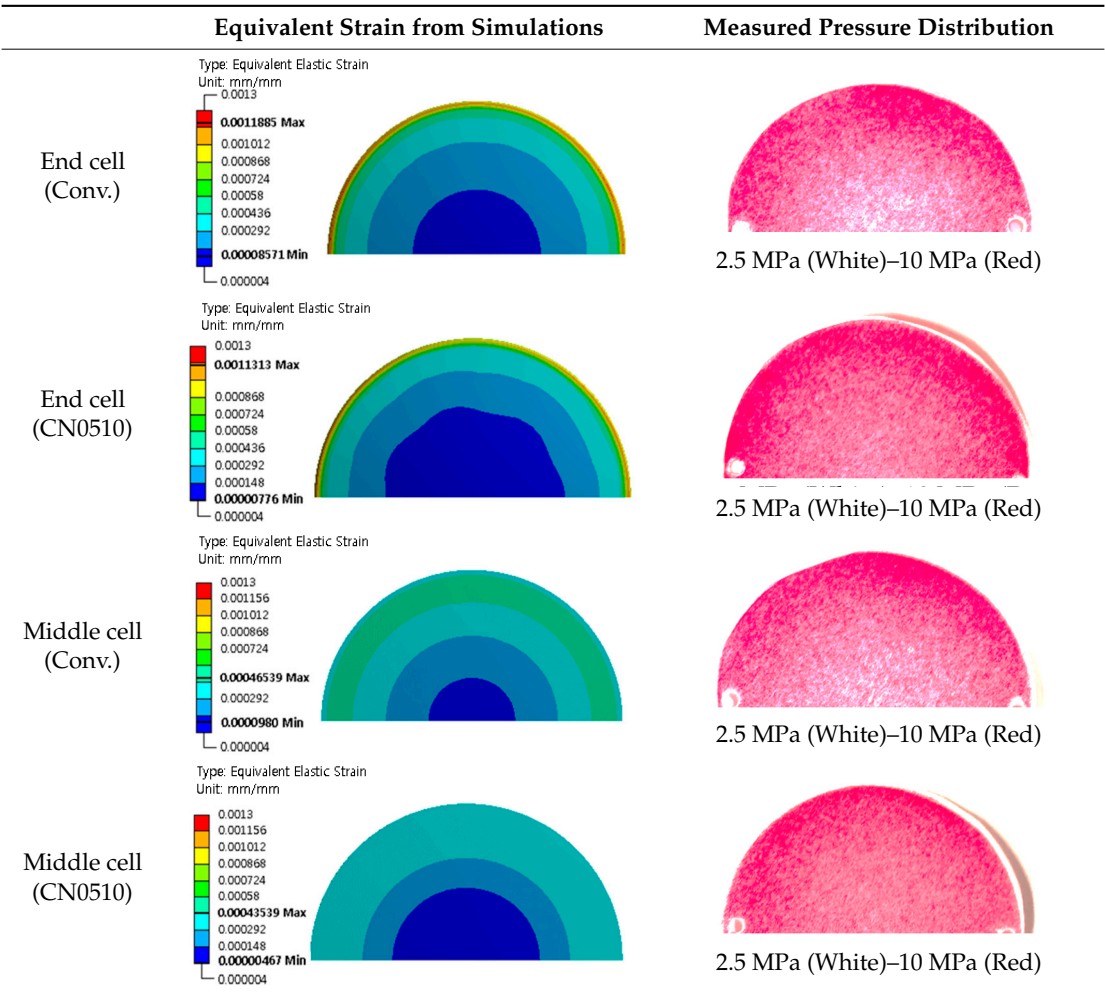

Furthermore, after comparing the cells located in the middle of the stacks to which the conventional circular end plate had been applied and those to which the CN0510 end plate had been applied, the strain was smaller, and the color was more uniform than for the end cells. The clamping pressure was uniformly applied to the end plates; thus, the CN0510 end plate and uniform clamping pressure were equivalent to that of a conventional circular end plate.

The strain of the end and the middle cells of the stacks in which the conventional square end plate and those in which the CN10 end plate, which was selected from the parametric study, were applied, can be confirmed from Table 6. The strain of the end and the middle cells of the stacks with the

conventional square end plates and those with the CN10 end plate are at the same level, while the strains of the square end plates are 10 times larger than those of the circular end plates. The outside of the cell was bent and appeared severely deformed similar to the circular end plates.

**Table 6.** Comparison of the simulated equivalent-strain distribution with the experimental pressure distribution, using the square end plates of the dummy-cell stack.

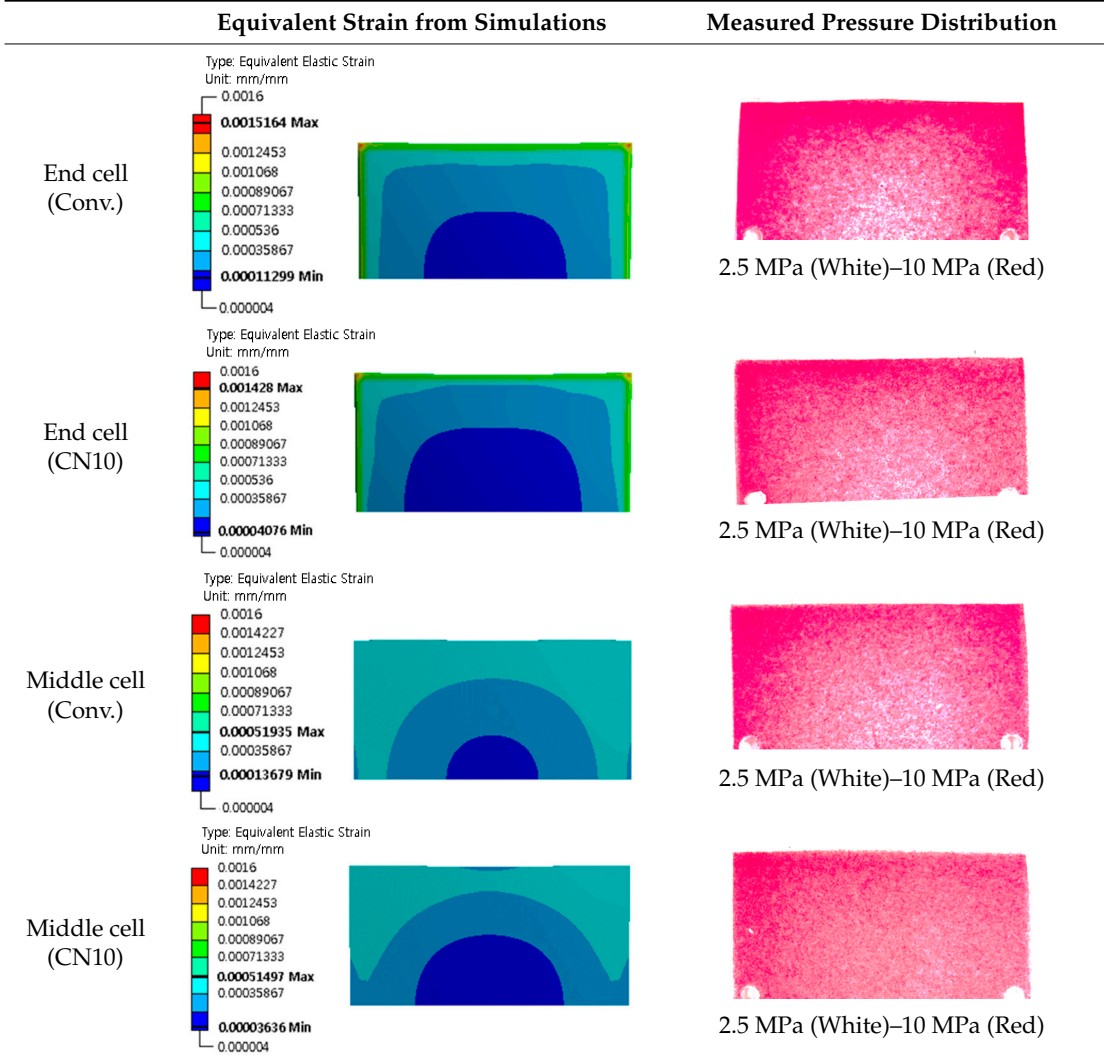

Furthermore, for cells located in the middle of the stacks with the conventional square end plate and those with the CN10 end plate, after comparing the end cells with rectangle contour inside, the strain was smaller, and the contour was circular. Evidently, the clamping pressure was applied uniformly, and so the square end plate with less weight and uniform clamping pressure was equivalent to that of the conventional square end plate. The contour distribution and color uniformity were confirmed by the experimental results with pressure sensing films.

Figure 7 is a graph comparing each of the four types of stacks in which conventional circular, CN0510, conventional square, and CN10 end plates had been applied. For the middle cell, the deviation of the strain is much smaller than at the end cells. The minimum strains of the middle cells are nearly zero, which can result in high ohmic resistance.

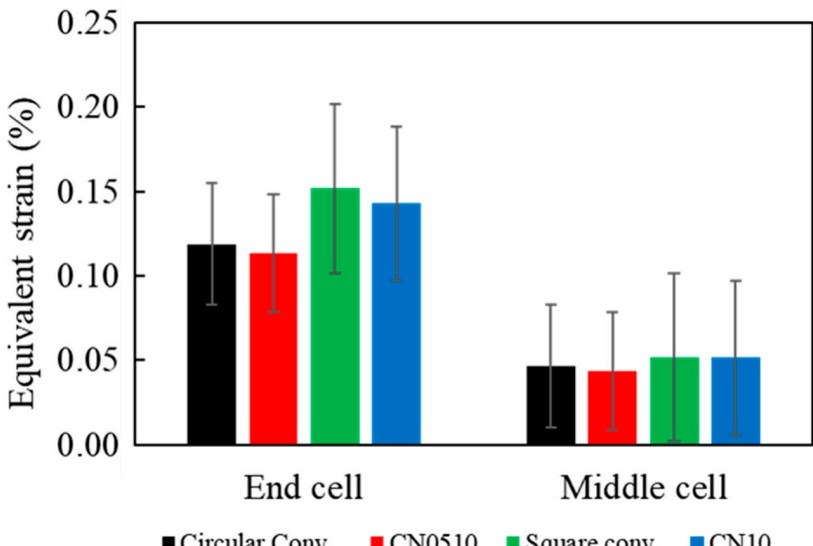

**Figure 7.** Strain deviation of middle and end cells of the stacks for conventional circular (black), CN0510 (red), conventional square (green), and CN10 (blue) end plates.

Figure 8 confirms the strain tendencies for each cell. In Figure 8, the stack applied with the conventional circular and CN0510 end plates had less strain at both ends of the cell compared with the stack with the conventional and CN10 end plates. The strains of the newly designed end plates (CN0510, CN10) are slightly lower than those of the conventional end plates. The cell strain is higher at both ends of the stack with all kinds of end plates.

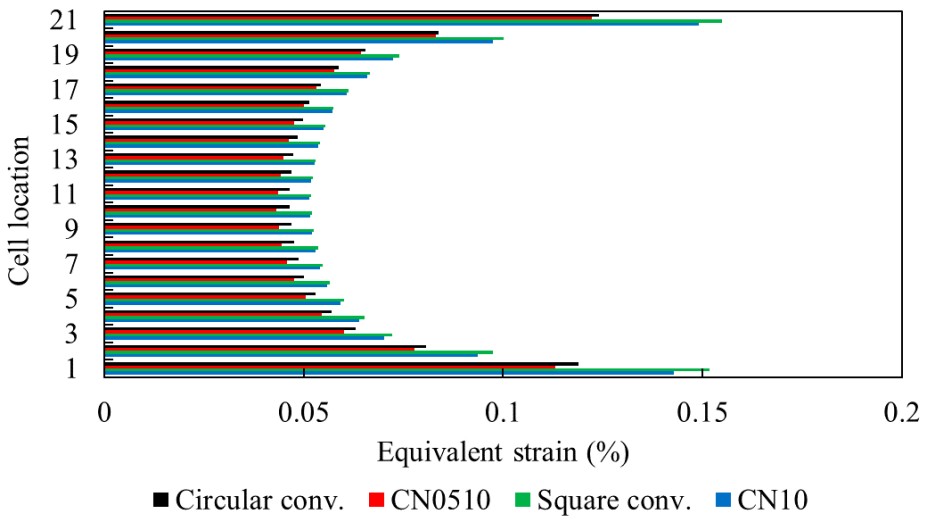

**Figure 8.** Maximum strain at each cell at the 21-cell stack.

## 5. Conclusions

Stress and deformation analysis and topology optimization were performed in this paper to solve the deterioration of PEMWE performance caused by deformation—a problem for the end plate—and reduce the weight of the two types of conventional circular and square end plates. Thus, light end plates were designed using the derived parameters. The weights of the light circular and square end plates were reduced compared with conventional end plates by 22.9% and 23.3%, respectively. Furthermore, strain analysis was performed to confirm the uniformity of the clamping pressure between cells. The stacks with the light circular and square end plates had confirmed strains similar to the stack applied with the conventional end plates.

Furthermore, even if the square end plates have large stress distributions compared with the circular end plates, the cell clamping pressure was distributed uniformly in each cell because of the nonlinearity of the square end plate pressure distribution. The results verified that structural strength was required and that the square end plates exhibited reduced strain due to the large stress concentration compared with the circular end plates. Thus, FEA was performed to examine the end plate weight reduction and the uniformity of the clamping pressure between the stack cells. The pressure sensing experiments validated simulation results qualitatively. The comparative analysis of the circular and the square end plates with dummy cell simulation can improve the performance of the high-pressure PEMWE by using end plates to provide gas-tight sealing and uniform clamping pressure.

**Author Contributions:** M.J. wrote the manuscript, conceptualized, and analyzed; H.-S.C. analyzed and validated results; and Y.N. analyzed and supervised. All authors have read and agreed to the published version of the manuscript.

**Funding:** This work was supported by the Basic Science Research Program through the National Research Foundation of Korea (NRF) funded by the Ministry of Science and ICT (grant number NRF-2019M3E6A1064703). The authors gratefully acknowledge partial financial support for this work by the Technology Development Program to Solve Climate Changes of the National Research Foundation (NRF) of Korea funded by the Ministry of Science and ICT (NRF-2015M1A2A2074657).

**Conflicts of Interest:** The authors declare no conflict of interest.

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
