# Peer review of "Comparative Analysis of Circular and Square End Plates for a Highly Pressurized Proton Exchange Membrane Water Electrolysis Stack"

_applsci, doi:10.3390/app10186315_

Round 1
Reviewer 1 Report
The research topic is interesting and was tackled using appropriate research methods.
Please review the level of English in the paper and also try to add clarity to some portions. As an example, maybe the beginning of line 152 should have been ”The designation of the end plates consists...”?
Author Response
We could have a valuable chance to correct our manuscript. The authors would like to deeply thank the reviewers for that. We have revised it according to the reviewers’ comments. Enclosed please find the revised manuscript. The following are the point to point answers for the comments from the reviewers.
Reviewer(s)' Comments to Author:
Referee: 1
Q1: The research topic is interesting and was tackled using appropriate research methods.
Please review the level of English in the paper and also try to add clarity to some portions. As an example, maybe the beginning of line 152 should have been “The designation of the end plates consists...”?
A1: We appreciate your positive comments of Reviewer #1. We have got the draft corrected by professional editorial company and attached the certification. Especially the expression you mentioned on line 152 was modified to line 171 as follows.
The designed end plate consists of six letters in total
àThe designation of the end plates consists of six letters.
Reviewer 2 Report
The manuscript presents ANSYS calculations of strain of different variants of end plates for electrolyzers.
Minor concerns
-
Fig. 1: poor contrast between the background and text in the images. Same in Figs. 2, 3, Table 3, 4, 5, 6,.
-
Fig. 3: Physical units are written in the caption. Writing it into the diagrams would make it more clear. Same in Table 3, 4.
-
Fig. 6: Physical units are missing. Same in Table 5, 6, Figs. 7, 8.
-
Fig. 8: Different font size of labels in Fig. 8 (a) and (b).
-
Not much literature was cited – only 13 references.
Major concerns
-
Page 6, lines 179-181: change of deformation was given in percent. How much deformation is acceptable? And why?
-
Validation is missing (see next point).
-
Table 6: the strain looks inhomogeneous along the edges. Does this possibly affect the tightness of the construction?
-
Table 6, strain distribution: From the described boundary conditions it could be expected that the strain should be symmetric in the x and y coordinate / axes through the center of the cross section shown in Table 6. Especially the strain along the edges loo asymmetric: the colors in the top left corners are different from the colors in the to right corners of the images. This asymmetry indicates inaccuracies in the calculation – they seem to be even larger than the discussed results in the conclusions (line 284 on page 15) - even though the strain distribution can merely be indicated by colors instead of numerical values. This leads to the final recommendation.
Author Response
We could have a valuable chance to correct our manuscript. The authors would like to deeply thank the reviewers for that. We have revised it according to the reviewers’ comments. Enclosed please find the revised manuscript. The following are the point to point answers for the comments from the reviewers.
Reviewer(s)' Comments to Author:
Referee: 2
Q1: The manuscript presents ANSYS calculations of strain of different variants of end plates for electrolyzers.
A1: We appreciate your positive comments of Reviewer #2.
- Minor concerns
Q2: Fig. 1: poor contrast between the background and text in the images. Same in Figs. 2, 3, Table 3, 4, 5, 6,.
A2: We appreciate valuable comments. We removed all the contrasts in the background of Fig, 1,2,3 and Table 3,4,5,6 as you mentioned.
Q3: Fig. 3: Physical units are written in the caption. Writing it into the diagrams would make it more clear. Same in Table 3, 4. Fig. 6: Physical units are missing. Same in Table 5, 6, Figs. 7, 8.
A3: Thank you for your comments. We added all the units that were only in the captions of Fig. 3, 6, 7, 8 and Table 3, 4, 5, 6 to the pictures.
Q4: Fig. 8: Different font size of labels in Fig. 8 (a) and (b).
A4: Thank you for your comments. We adjusted the font size of Fig 8.
Q5: Not much literature was cited – only 13 references.
A5: We appreciate your positive comments. According to the Reviewer’s comments, we cited 20 references in total. Following 7 literatures are added.
- Al Shakhshir, S.; Frensch, S.; Kær, S.K. On the Experimental Investigation of the Clamping Pressure Effects on the Proton Exchange Membrane Water Electrolyser Cell Performance. ECS Trans. 2017, 77(11), 1409.
- Montanini, R.; Squadrito, G.; Giacoppo, G. Experimental evaluation of the clamping pressure distribution in a PEM fuel cell using matrix-based piezoresistive thin-film sensors. XIX IMEKO World Congress Fundamental and Applied Metrology, Lisbon, Portugal. 06-11 Sep. 2009.
- Al Shakhshir, S.; Zhou F.; Kær, S.K. On the Effect of Bipolar Plate Mechanical Properties on the Current Distribution of Proton Exchange Membrane Water Electrolysis. ECS Trans., 2018, 86(13), 683.
- Al Shakhshir, S.; Zhou F.; Kær, S.K. On the Effect of Clamping Pressure and Methods on the Current Distribution of a Proton Exchange Membrane Water Electrolyzer. ECS Trans., 2018, 85(13), 995.
- Immerz, C.; Schweins, M.; Trinke, P.; Bensmann, B.; Paidar, M.; Bystroň, T.; Bouzek, K.; Hanke-Rauschenbach, R. Experimental characterization of inhomogeneity in current density and temperature distribution along a single-channel PEM water electrolysis cell. Electrochim. Acta 2018, 260, 582-588.
- Verdin, B.; Fouda-Onana, F.; Germe, S.; Serre, G.; Jacques, P. A.; Millet, P. Operando current mapping on PEM water electrolysis cells. Influence of mechanical stress. Int. J. Hydrogen Energy 2017, 42(41), 25848-25859.
- Selamet, Ö. F.; Becerikli, F.; Mat, M. D.; Kaplan, Y. Development and testing of a highly efficient proton exchange membrane (PEM) electrolyzer stack. Int. J. Hydrogen Energy 2011, 36(17), 11480-11487.
- Major concerns
Q6: Page 6, lines 179-181: change of deformation was given in percent. How much deformation is acceptable? And why?
A6: Expressing the deformation in percentage is for the relative comparison, and the acceptable absolute value varies depending on what the stack consists of, such as MEA, GDL, and sealant. So, regardless of which internal components are used, this study conducted a qualitative comparison of how much internal deformation would be caused by a simple deformation of the end plates.
Q7: Validation is missing (see next point).
A7: Thank you for your good suggestion. Accordingly, we conducted experiments with dummy cells and pressure sensing films, and we verified it with a 1/4 length-scale geometry for qualitative verification. We identified that the internal stress was relatively less than external, and that the middle cell was more even than the end cell, so we were able to secure qualitative verification of the simulation.
Q8: Table 6: the strain looks inhomogeneous along the edges. Does this possibly affect the tightness of the construction?
Table 6, strain distribution: From the described boundary conditions it could be expected that the strain should be symmetric in the x and y coordinate / axes through the center of the cross section shown in Table 6. Especially the strain along the edges loo asymmetric: the colors in the top left corners are different from the colors in the to right corners of the images. This asymmetry indicates inaccuracies in the calculation – they seem to be even larger than the discussed results in the conclusions (line 284 on page 15) - even though the strain distribution can merely be indicated by colors instead of numerical values. This leads to the final recommendation.
A8: Thank you for your comment. In order to eliminate the model’s unevenness, the more mesh was used to perform the simulation, and we were able to achieve symmetric results and validated. Thank you for your advice.
Reviewer 3 Report
This paper discusses about improvement of the performance of the high pressure PEMWE by using end plates that can provide gas tight sealing and uniform clamping pressure based on the finite element analysis and topology optimization results. The manuscript is well written and the results are considered valuable for researchers in this area. However, some information should be added to improve the quality of the manuscript. My comments are listed below.
- Dimensions of the end plates should be provided. Is there any difference in plate area between circular and square end plates? If yes, how did the difference affects the stress and deformation of the end plates?
- The number of bolts between circular and square end plates is not equal. I am curious that the number of bolts used in the end plates changes the stiffness of the whole PEMWE stack. A parameter such as plate area per bolt might be useful to understand the difference. Is there any comment on this matter?
- Stress and strain used in this study seemed to be equivalent stress and strain which are scalar values. Did you also look at the principal and shear stresses (strains)? It would be good if authors can explain what kind of deformation mode occurs in the end plates and in the stack.
- According to the topology optimization result, end plates with complicated shapes were obtained. However, a lot of cost will be needed to manufacture the end plates compared to the conventional end plates. Do you think it is still worth to manufacture the optimized end plates? Is there any other benefit of weight reduction instead of lowering unnecessary material consumption as mentioned in Introduction?
Author Response
We could have a valuable chance to correct our manuscript. The authors would like to deeply thank the reviewers for that. We have revised it according to the reviewers’ comments. Enclosed please find the revised manuscript. The following are the point to point answers for the comments from the reviewers. Please see the attachment.

Round 2
Reviewer 3 Report
Axis label of Figure 6 to Figure 8: "strain" should be written as "equivalent strain".
Author Response
The authors would like to deeply thank the reviewers for revision. We have revised it according to the reviewers’ comments. Enclosed please find the revised manuscript.
Reviewer(s)' Comments to Author:
Referee: 1
Q1: Axis label of Figure 6 to Figure 8: "strain" should be written as "equivalent strain".
A1: We appreciate your positive comments of Reviewer #1. We changed the axis label of Figure 6, 7, 8 to “Equivalent strain (%) as follows:
